

# Genome-wide identification and characterization of GRAS transcription factors in tomato (*Solanum lycopersicum*)

Yiling Niu, Tingting Zhao, Xiangyang Xu and Jingfu Li

College of Horticulture and Landscape, Northeast Agricultural University, Harbin, China

## ABSTRACT

*Solanum lycopersicum*, belonging to Solanaceae, is one of the commonly used model plants. The GRAS genes are transcriptional regulators, which play a significant role in plant growth and development, and the functions of several GRAS genes have been recognized, such as, axillary shoot meristem formation, radial root patterning, phytohormones (gibberellins) signal transduction, light signaling, and abiotic/biotic stress; however, only a few of these were identified and functionally characterized. In this study, a gene family was analyzed comprehensively with respect to phylogeny, gene structure, chromosomal localization, and expression pattern; the 54 GRAS members were screened from tomato by bioinformatics for the first time. The GRAS genes among tomato, *Arabidopsis*, rice, and grapevine were rebuilt to form a phylogenomic tree, which was divided into ten groups according to the previous classification of *Arabidopsis* and rice. A multiple sequence alignment exhibited the typical GRAS domain and conserved motifs similar to other gene families. Both the segmental and tandem duplications contributed significantly to the expansion and evolution of the GRAS gene family in tomato; the expression patterns across a variety of tissues and biotic conditions revealed potentially different functions of GRAS genes in tomato development and stress responses. Altogether, this study provides valuable information and robust candidate genes for future functional analysis for improving the resistance of tomato growth.

Corresponding author
Xiangyang Xu, nxyyyx@neau.edu.cn, xxy709@126.com

## INTRODUCTION

Transcription factors comprise the core of functional genomics. These transcription factors are protein molecules that can either activate or repress the target genes to ensure their specific expression by combining with the genes of 5′-flank cis-element (*Riechmann et al., 2000*). The typical transcription factor consists of DNA-binding domain, transcription regulation domain (active region or suppressor region), oligomerization site, and nuclear localization signal (*Morohashi et al., 2003*). GRAS is a major plant-specific transcription factor gene family among putative transcription factors that are found in a variety of plant species; for example, *Arabidopsis (Arabidopsis thaliana)*, rice (*Sativa oryza*), grapevine (*Vitis vinifera*), tobacco (*Nicotiana tabacum*), Chinese cabbage (*Brassica rapa ssp. pekinensis*), and *Prunus mume*. Based on the known first three members, GAI (gibberellic acid insensitive)

(*Peng et al., 1997*), RGA (repressor of GA1-3 mutant (*Silverstone, Ciampaglio & Sun, 1998*), and SCR (scarecrow) (*Di Laurenzio et al., 1996*), this transcription factor was named GRAS with the characteristic letter from each of the three members. The family members were screened as GRAS gene family as each of them contained the GRAS domain (*Pysh et al., 1999*). The GRAS proteins are usually composed of 400–700 amino acid residues (*Bolle, 2004*). A few GRAS proteins contain two structural domains: one GRAS domain and the other functional domain (*Schumacher et al., 1999*). The typical features of these proteins include a highly conserved C-terminal region and a variable N-terminal region (*Sun et al., 2011*). The conserved C-terminal region harbors five sequence motifs: LHRI (leucine heptad repeat I), LHRII (leucine heptad repeat II), VHIID motif (*Pysh et al., 1999*), SAW motif, and PFYRE motif. The structures of both LHR I and LHR II consitute two leucine enrichment regions, the VHIID motif is a core structure, which exists in all members of the GRAS gene family and it can combine with LHR I and II to form the complex LHR I-VHIID-LHR II. This structural pattern might be a crucial function for DNA-binding and protein-binding in the protein–protein interactions (*Itoh et al., 2002*). The localization of SAW and PFYRE motifs, for functional specificity, is not yet clearly elucidated; however, the missense mutations in these motifs in RGA and SLR1 proteins exhibit strong mutant phenotypes (*Silverstone, Ciampaglio & Sun, 1998*). In addition, the N-terminal region of GRAS proteins in the DELLA subfamily contains the other two motifs: DELLA and VHYNP, and the VHYNP motif is dynamic, implying that the N-terminal domains of GRAS proteins harbor various motifs (*Peng et al., 1997*). Moreover, the GRAS proteins are not only structurally diverse but also exert multiple functions. In recent years, several groups have found that the GRAS proteins are one of the indispensable regulatory factors in plant growth development and participate in several biochemical and physiological processes in plants, such as gibberellin signal transduction (*Peng et al., 1997*; *Silverstone, Ciampaglio & Sun, 1998*; *Ikeda et al., 2001*), axillary shoot meristem formation (*Stuurman, Jaggi & Kuhlemeier, 2002*), root radial patterning (*Di Laurenzio et al., 1996*; *Helariutta et al., 2000*), male gametogenesis (*Morohashi et al., 2003*), phytochrome A signal transduction (*Bolle, Koncz & Chua, 2000*), nodulation signal transduction (*Hirsch et al., 2009*), and biotic/abiotic stress (*Huang et al., 2015*). Previously, the GRAS proteins were divided into eight subfamilies, according to their common feature or member; for instance, SHR, SCR, DELLA, LISCL, Ls, HAM, PAT1, and SCL3. The amino acids in each group are mostly homologous, and thus, the GRAS genes of each subfamily might possess similar or related functions (*Tian et al., 2004*).

Hitherto, the RNA-seq is developed to identify the specific genes of the GRAS family; subsequently, the functions of these genes are investigated. Some GRAS genes have been characterized based on *Arabidopsis* and rice; for example, the DELLA proteins that function as negative regulators in gibberellin signal transduction, the GA signaling pathway regulates the plant growth and development by degrading the DELLA proteins (*Zhang et al., 2011*; *Heo et al., 2011*). Firstly, the signal perception of GID1 proteins receive the GA signal; then the GID1 proteins combine with the DELLA proteins to form the complexes of GA-GID1-DELLA. Subsequently, the DELLA proteins specifically bind to the F-box protein SLY1, which subordinates the SCF$^{SLY1/GID2/SNE}$ protein complex. Finally, the degradation of the

DELLA proteins mediated by the 26S proteasome released the inhibition, and thus, the plants showed a normal growth (*Day et al., 2004*). The study showed that the DELLA motif in the N-terminal is indispensable for the interaction between the two proteins DELLA and GID1; however, the motifs in the C-terminal are redundant. The SCR and SHR groups and the analysis of SHR/SCR mutant showed a short root phenotype (*Di Laurenzio et al., 1996*; *Helariutta et al., 2000*), providing evidence that both proteins act as positive regulators in the radial organization of the root. Previous studies have shown that the SCR proteins combine with the SHR proteins to form a complex, while the SHR proteins were transported to the endodermis of the root (*Sabatini et al., 2003*; *Wysocka-Diller et al., 2000*; *Helariutta et al., 2000*). Similarly, SCL3 proteins were involved in the elongation and differentiation region of the root and over-ground organs, respectively. In the root, the subsequent elongation is regulated to control the GA signaling pathway, while in the meristematic tissue, the combination of SHR/SCR leads to the organizational maturity within the GA signaling pathway (*Heo et al., 2011*). Some studies indicate that the overexpression of *OsMOC1* gene results in increasing the tiller numbers and decreasing the length; moreover, it can only be detected in the axillary bud (*Li et al., 2003*; *Li, Stoeckert Jr & Roos, 2003*). In addition to rice, the *Ls* gene in tomato and the *AtLAS/SCL18* gene in *Arabidopsis* are closely linked to the growth of the lateral bud (*Schumacher et al., 1999*; *Greb et al., 2003*). The *PAT1*, *SCL13*, and *SCL21* genes belong to the *PAT1* branch, which mediates the phytochrome signaling pathways. Genetic evidence suggests that three genes act as positive regulators; the *SCL13* gene participates in phytochrome B transduction independently, whereas the*PAT1* and *SCL21* adjust the phytochrome A signaling network by the mutual effect (*Bolle, Koncz & Chua, 2000*; *Torres-Galea et al., 2006*; *Torres-Galea, Hirtreiter & Bolle, 2013*). The *NSP1* and *NSP2* genes exist in the downstream of symbiosis signal transduction pathway CCaMK (Ca/calmodulin-dependent protein kinases), which is related to the nodulation. TheNPS1-NPS2 heterodimer is induced by the nodulation factors, following which, the heterodimer can specifically combine the promoter of the early nodulation *ENOD2* gene to promote the expression of related genes, thereby forming the stage of nodulation (*Hirsch et al., 2009*). The homologous genes of *NSP1/NSP2* are extensively encompassed in several non-leguminous plants. Another analysis about GRAS proteins found that some members in the gene family are regulated by miRNA171; for instance, *AT2G45160*, *AT3G60630*, and *AT4G00150* in *Arabidopsis* (*Schulze et al., 2010*), *Pm017821* and *Pm023512* in *Prunus mume* (*Wang et al., 2014*), *Solyc01g090950.2.1* and *Solyc08g078800.1.1* in tomato (*Huang et al., 2015*), and four genes in rice (*Llave et al., 2002*) are complementary to miRNA171.

In the past few years, some investigations showed that the GRAS genes respond to different hormones and abiotic stress treatments (*Huang et al., 2015*). In this study, the 53 GRAS genes, the phylogenetic tree, and the expression patterns in different abiotic stress treatment were investigated; however, only 48 tomato GRAS genes were selected for phylogenetic analysis. The abiotic stress analysis focused on salt, cold, hat, and osmotic and drought stress; only about 40 tomato GRAS genes were investigated, and the analysis of bioinformatics was poor. This study selected a large number of tomato GRAS genes for several bioinformatic analyses, and analyzed the relationship between pstDC3000 and

GRAS genes from tomato for the first time. In this study, a comprehensive and systematic analysis would provide a deep insight on the GRAS family and precede the sequencing studies. The tomato is an adequate model plant for the study of Solanaceae due to its great economic value. With the whole-genome analyses of the tomato, the genomic data can highlight the connection between genes and plants.

## MATERIALS & METHODS

### Identification of tomato GRAS genes

We retrieved the tomato GRAS genome sequences, protein sequences, and annotation information from SGN (https://solgenomics.net/) (*The Tomato Genome Consortium, 2012*). The Arabidopsis GRAS genes' family sequences and annotation information was downloaded from TAIR (http://www.arabidopsis.org/) (*Swarbreck et al., 2008*), whereas the rice GRAS transcription factor sequences and annotation information was obtained from RGAP (http://rice.plantbiology.msu.edu/index.shtml) (*Ouyang et al., 2007*). The grape GRAS gene family sequences were downloaded from plantTFDB (http://planttfdb.cbi.pku.edu.cn/). The HMM model of GRAS transcription factor was downloaded from PFAM (http://pfam.xfam.org/) (*Finn et al., 2010*), known as PF03514. The HMM model was used as a query to identify the tomato GRAS genes containing the GRAS domain with a cut-off $E$-value of $1e^{-5}$ in the HMMER software. Consequently, we identified a total of 54 GRAS genes in tomato, 34 from Arabidopsis, 56 from rice, and 37 from the grapevine. Then, we conducted a quality check using the Simple Modular Architecture Research Tool SMART (http://smart.embl-heidelberg.de/) (*Letunic, Doerks & Bork, 2012*) to confirm the presence of GRAS domain on the candidate GRAS genes. These data were used for subsequent analysis.

### Phylogenetic analysis for tomato GRAS genes

MEGA 6.0 was used to analyze the phylogenetics of genome-wide GRAS gene family based on the whole set of GRAS protein sequences from tomato (*solanum lycopersicum*), Arabidopsis (*Arabidopsis thaliana*), rice (*Sativa oryza*), and grapevine (*Vitis vinifera*). The *Arabidopsis* is one of the most commonly used plants in Cruciferae for studying the genetic correlations; *Arabidopsis* to *Cruciferae*, grapevine to *Vitaceae*, and rice to Gramineae (*Tamura et al., 2011*). In this building process, several shorter amino acids (*Solyc01g090950.1.1, Solyc04g011630.1.1, Solyc12g049320.1.1*) were excluded, the domain length of these sequences were shorter than half of the typical GRAS domain (350 amino acids) with low similarity among the tomato GRAS family (*Huang et al., 2015*). According to a previous study, the classification was made on the phylogenetic tree using the Evolview software (http://www.evolgenius.info/).

### Structure analysis of tomato GRAS genes

Using the NCBI platform (https://www.ncbi.nlm.nih.gov/), the conserved domains of 54 GRAS genes were visualized. In order to search for the potentially conserved motifs in the complete amino acid sequence of tomato GRAS proteins, the Multiple EM for Motif Elicitation (http://meme-suite.org/) (*Bailey et al., 2006*) was used with default parameters,

except that the number of motifs was set to 10. In order to present the characteristic of every subfamily, the 54 GRAS genes were used to build a Maximum Likelihood tree based on the JTT matrix-based model. Other parameters were same as above. The secondary structures of the tomato GRAS genes were generated using the Gene Structure Display Sever (http://gsds.cbi.pku.edu.cn/) (Guo et al., 2007).

## Evolution analysis of tomato GRAS genes

The sequences were compared using the GRAS genes in tomato, *Arabidopsis*, and rice; the entire protein sequences were used to identify the orthologous and paralogous genes using the software OrthoMCL (http://orthomcl.org/orthomcl/) with an *E*-value of $1e^{-5}$ and a match cut-off value 50 for against all BLASTp alignments (Li, Stoeckert Jr & Roos, 2003; Li et al., 2003). The MCscanX software (Tang et al., 2008) was used to identify the collinear block based on the tomato genomes; if a gene had more than one transcript, only the first transcript in the annotation was used. The underlying mechanism of tandem duplication showed that the two genes were physically close to each other with the genes residing within 20 kb (Liu et al., 2014), segmental duplication resulted from the whole genome duplication accompanied by a comprehensive gene loss (Tang et al., 2008), and those large duplication events can be deduced by anchor genes in collinear blocks (Cannon et al., 2004). The chromosomal localization and homologous collinear relationship of GRAS genes were visualized using the Circos program (Krzywinski et al., 2009).

## Expression pattern analysis of GRAS genes

We utilized the Illumina RNA-seq data of tomato download from SGN (https://solgenomics.net/), which were reported previously. To confirm the expression patterns of the GRAS genes, the FPKM was used to represent the expression level of each tomato gene. The data on GRAS genes' expression was retrieved to display the consequences using the HemI program (http://hemi.biocuckoo.org/); the expression data were amplified 100-fold (Deng et al., 2014). The transcriptomic data were extracted from twelve tissues of *S. pimpinellifolium* (LA1589), and *S. pimpinellifolium* was abundant in genetic variation, which can better assess the evolution of the GRAS genes. The organism comprised of the following: A, newly developed leaves approximately 5 mm long; B, mature green leaflets; C, flower buds 10 days before anthesis or younger; D, flowers at anthesis (0DPA); E, 10 days post-anthesis fruit (10DPA); F, 20 days post-anthesis fruit (20 DPA); G, breaker stage ripening fruit (33DPA); H, another set of 10DPA fruit that was collected in a separate greenhouse for comparison. The following tissues were collected from seeds that were germinated and grown for 7 days in a Petri dish under glowing lights; I, whole root; J, hypocotyl from below the cotyledons to above the root zone; K, cotyledons; L, vegetative meristems.

In the case of difficulty, other public data were retrieved to reveal the relationship between GRAS genes and biotic stress. The transcriptome of leaves of resistant (RG-PtoR) and susceptible (RG-prf3 and RG-prf19) tomato plants treated with pstDC3000 in 4 and 6 h were sequenced.

## RESULTS

### Genome-wide identification and annotation of the GRAS genes in tomato

In order to identify the GRAS proteins in tomato, we downloaded the raw data sequence of tomato GRAS transcription factors from the SGN database. The bioinformatics approach retrieved 54 tomato GRAS genes, 34 *Arabidopsis* GRAS genes, 56 rice GRAS genes, and 37 grape GRAS genes. In addition, we obtained basic information in connection with the tomato GRAS proteins (Table S1). The length of GRAS proteins ranged from 125–864 aa and the tomato GRAS genes were almost distributed across the 12 chromosomes uniformly; the highest content was on chromosome 1, containing eight GRAS genes. However, at least three GRAS genes were located on the other chromosomes. The annotation information represented that the GRAS domain always assembled on the C-terminal, suggesting that the conserved terminal was supported by the stable structure of the GRAS domain; this phenomenon could be used for the subsequent comparison analysis. The smaller HMM *E*-value provided reliable screening results. Notably, a number of amino acid residues in some GRAS proteins were less than that in the typical GRAS domain, such as *Solyc04g011630.1.1* or *Solyc06g076290.1.1* genes.

### Genome-wide sequence alignment and phylogenetic analysis

The topology tree contains 178 proteins: 56 from rice, 34 from *Arabidopsis*, 37 from grapevine, and 51 from tomato. However, the remaining three proteins from tomato were neglected because of the shorter domain sequence. Basing on the topology structure, clade support value, and previous classification from rice and *Arabidopsis*, as shown in Fig. 1 (*Tian et al., 2004*), all of these proteins were divided into ten groups, named specifically after a common feature or one of the members: PAT1, SHR, LISCL, Gv6, SCR, LAS, SCL3, DELLA, HAM, and BolA, respectively. Of these, eight were named according to a previous report (*Tian et al., 2004*). The remaining two groups were named by the characteristic of the members. The tree showed that the four species GRAS proteins were randomly distributed in the ten groups, and hence, the members of GRAS transcription factors in these four species are not represented equally. For example, the subfamily SCL3 contains one or two proteins from *Arabidopsis*, tomato, and grapevine, respectively, and seven members from rice. Strikingly, some clades do not contain the members of GRAS proteins from *Arabidopsis* or rice; for instance, Gv6 and BolA groups. This might be attributed to the loss of the corresponding member following the separation of the last common ancestor if not the problem of assembling or annotation of the *Arabidopsis* or rice genomes. We found that the Gv6 group consisted of six tomato proteins and two grapevine proteins, indicating that it is a fruit tree species-specific clade.

The genes with orthologs frequently tend to be clustered together, and the subfamily members in a major group share similar gene structure and function. Therefore, we observed that the distribution of specific genes provides information on the role of the other genes in the same clade. For example, within the PAT1 group, *AT5G48150.1* of *Arabidopsis* was previously demonstrated to participate in the process of phytochrome signal transduction. In addition, the *Solyc07g047950.1.1* was highly similar to the *AT5G48150.1*

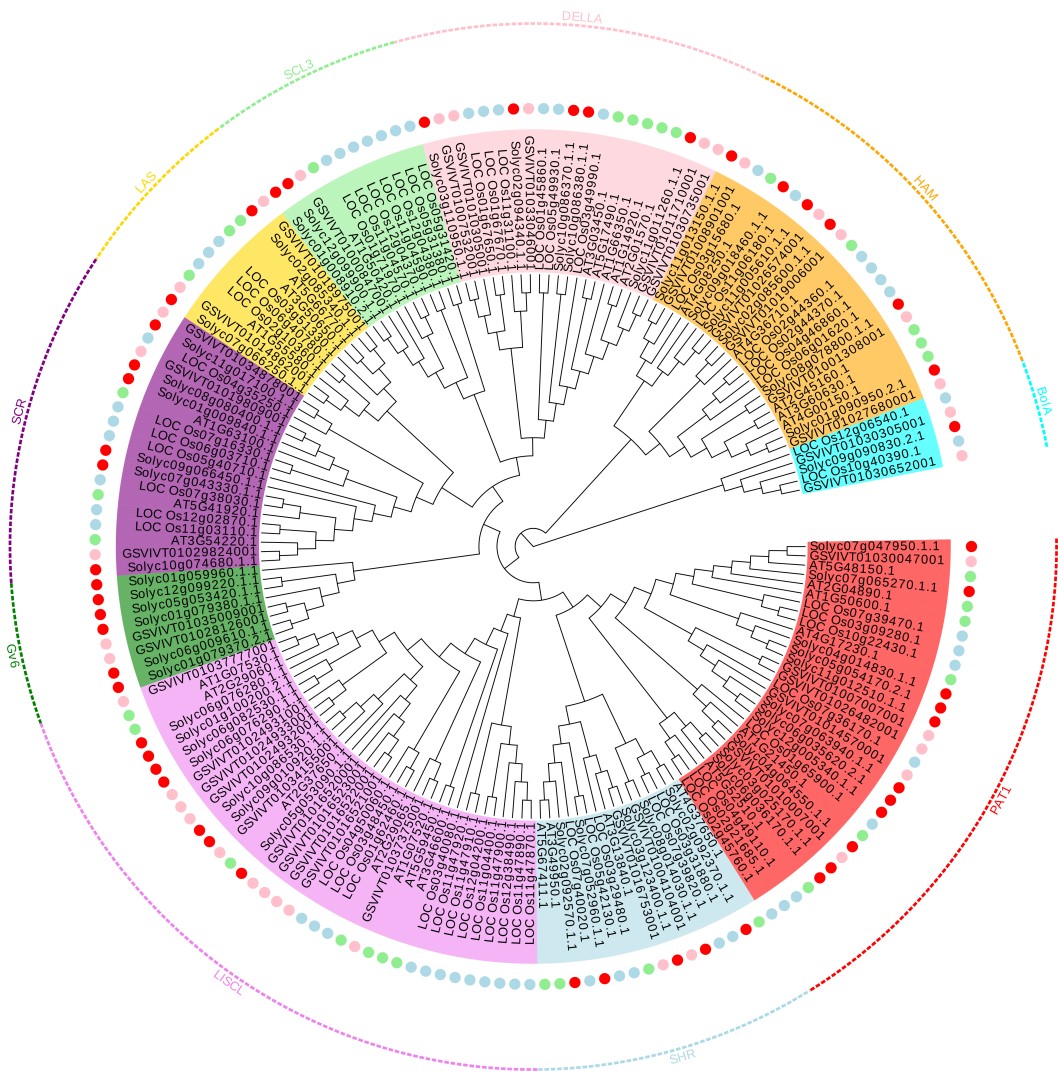

**Figure 1** **Phylogenetic tree of GRAS proteins from tomato, *Arabidopsis*, rice and grapevine, respectively.** Members in the same sub-branch were marked by the same color and surrounded with an appropriate color line that was commanded by the subfamily name. The red circle corresponds to the tomato GRAS proteins, the green circle corresponds to the *Arabidopsis* GRAS proteins, the blue circle corresponds to the rice GRAS proteins, and the pink circle corresponds to the grapevine GRAS proteins.

protein, and hence, we inferred that *Solyc07g047950.1.1* protein also played a crucial role in phytochrome signal transduction. PAT1 and LISCL subfamily comprised of more proteins than the other subfamilies as well as that reported in previous studies.

Members in the same sub-branch were marked by the same color and surrounded with an appropriate color line that was commanded by the subfamily name. The red circle corresponds to the tomato GRAS proteins, the green circle corresponds to the *Arabidopsis* GRAS proteins, the blue circle corresponds to the rice GRAS proteins, and the pink circle corresponds to the grapevine GRAS proteins.
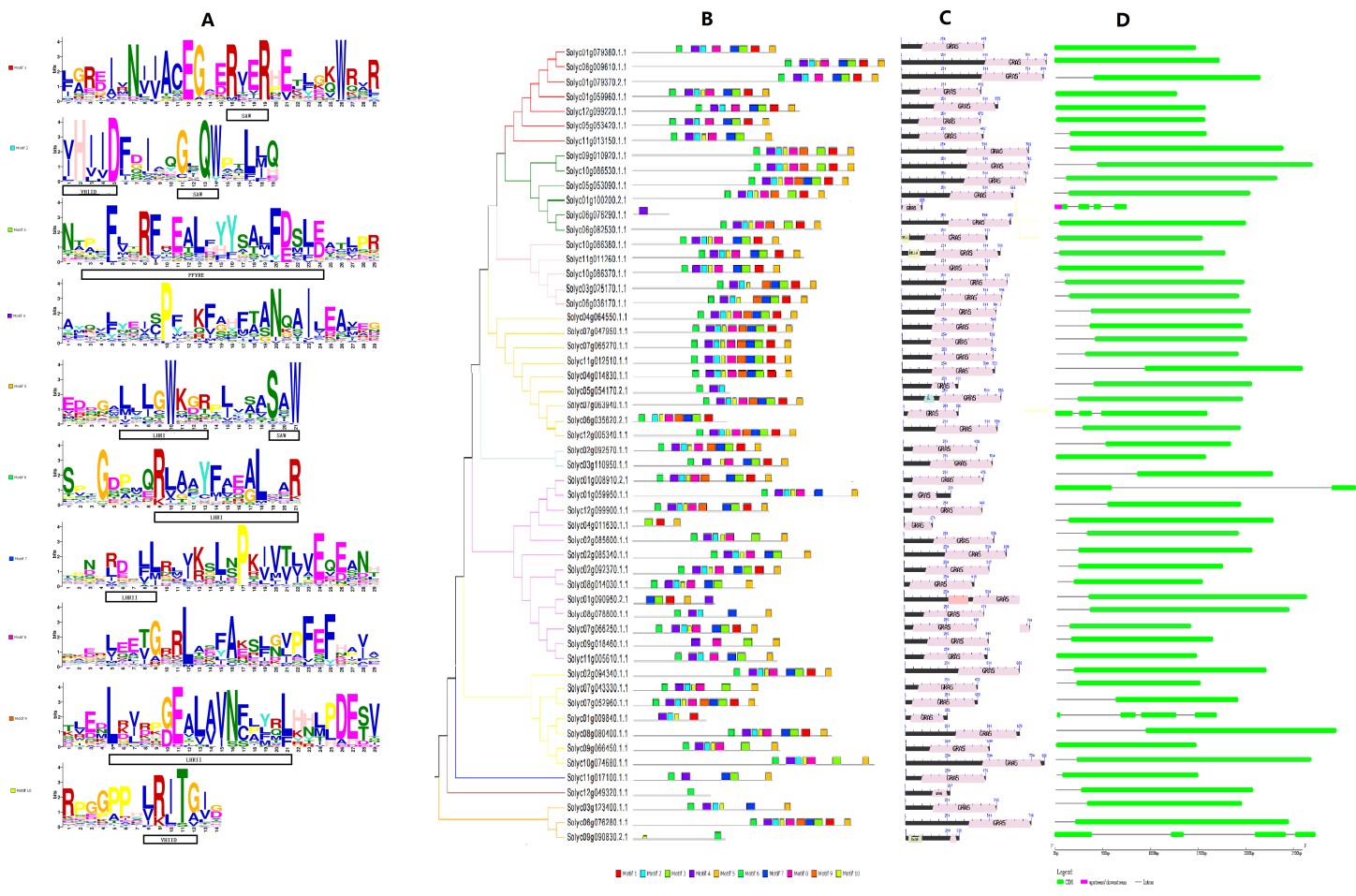

**Figure 2** **The structure of GRAS genes in tomato.** (A) Motif logo, the amino acid composition of each motif; (B) the motif distribution in each GRAS gene; (C) the location of GRAS domain in genes; (D) the length of exons, introns and UTRs.

## Structural analysis of GRAS genes (motif analysis and exon/intron analysis)

To achieve a general overview of the conserved features of the tomato GRAS proteins, we performed a multiple sequence alignment of the 54 tomato GRAS genes using MEME software; the conserved motifs of all proteins are shown in Fig. 2B. Ten default motifs were identified by MEME analysis. We found that 80% GRAS proteins contained >7 motifs, and 24% GRAS proteins encompassed 10 motifs. Moreover, members of the same clade of phylogenetic tree shared similar motif organization with respect to either gene length or motif number (*Liu et al., 2014*). On the other hand, several proteins only contained one or two motifs, which might be attributed to the short duration, resulting in a shorter domain. Also, the motifs were more likely to be located in the C-terminal than the N-terminal. The motif logo is shown in Fig. 2A. According to the previous studies on GRAS motifs (*Tian et al., 2004*), the five putative GRAS domains were exhibited in the logo of motifs; we speculated that each LHRI, LHRII, VHIID, PFYRE, and SAW motifs could be divided

into different units. This phenomenon demonstrated that maximum tomato GRAS genes are conserved, and the conserved region is localized on the C-terminal. Furthermore, we conducted a domain analysis (Fig. 2C) and exon/intron analysis (Fig. 2D) of tomato GRAS genes in order to explore the diversity and functionality. The visualization of the domains revealed that nearly all the GRAS proteins domains were primarily distributed in the C-terminal, further supporting the theory that the C-terminal is highly conserved. Nevertheless, there were some specific genes, such as the whole *Solyc06g076290.1.1* gene has one GRAS domain, *Solyc10g086380.1.1*, *Solyc11g011260.1.1*, *Solyc07g063940.1.1*, *Solyc01g090950.2.1*, and *Solyc09g090830.2.2* genes contained two domains. In addition, the exon/intron analysis displayed similar distribution characteristics. The *Solyc06g076290.1.1* and *Solyc090830.2.2* genes are comprised of multiple exons; however, the majority of GRAS proteins possessed only one exon and one intron, which might result from intron gain or loss event during evolution.

## Comparison and expansion analysis of tomato GRAS genes

To evaluate the evolutionary relationships among tomato GRAS genes, we made a comparative analysis to identify the co-orthologous, orthologous, and paralogous GRAS genes among tomato, *Arabidopsis*, and rice using the OrthoMCL software, as shown in Fig. 3. We identified 22 co-orthologous and 46 orthologous gene pairs between tomato and *Arabidopsis*, 45 co-orthologous and 48 orthologous gene pairs between tomato and rice, and 19 co-orthologous and 29 orthologous gene pairs between *Arabidopsis* and rice. Moreover, 38 tomato GRAS genes (65%) have paralogous genes, which was higher than that in rice (63%) and *Arabidopsis* (19–56%). The orthologous genes commonly share a similar structure and biological function. The number of orthologous gene pairs between tomato and rice was more than that between tomato and *Arabidopsis*, which indicated that tomato is similar to rice.

Moreover, the duplication events were discovered in the evolution of tomato GRAS transcription factors (*Cannon et al., 2004*). Tandem and segmental duplication were vital for the expansion of the GRAS family. To reveal the relationship between gene duplication and amplification, the syntenic regions were analyzed by MCscanX software. As shown in Fig. 4, finally, we obtained five tandem duplication gene pairs (18.5%) (the details of the duplication events were summarized in Table S2), and the result suggested that the origination of GRAS genes applied to the tandem duplication events. We also assessed the contribution of segmental duplications; 18 genes (33.3%) with duplications were harbored in collinear blocks, indicating their robust participation in the expansion of GRAS genes. The same observations were also made in other protein families (*Wang et al., 2016*).

## Analysis of expression pattern tomato GRAS genes in different tissues

In this study, we analyzed different expression levels of GRAS proteins in various tissues regarding the published RNA-seq data. The heat map was constructed to show the expression profile and the genes listed according to their branch (Fig. 5). We found that only the *Solyc01g059960.1.1* gene was not detected in the RNA-seq data

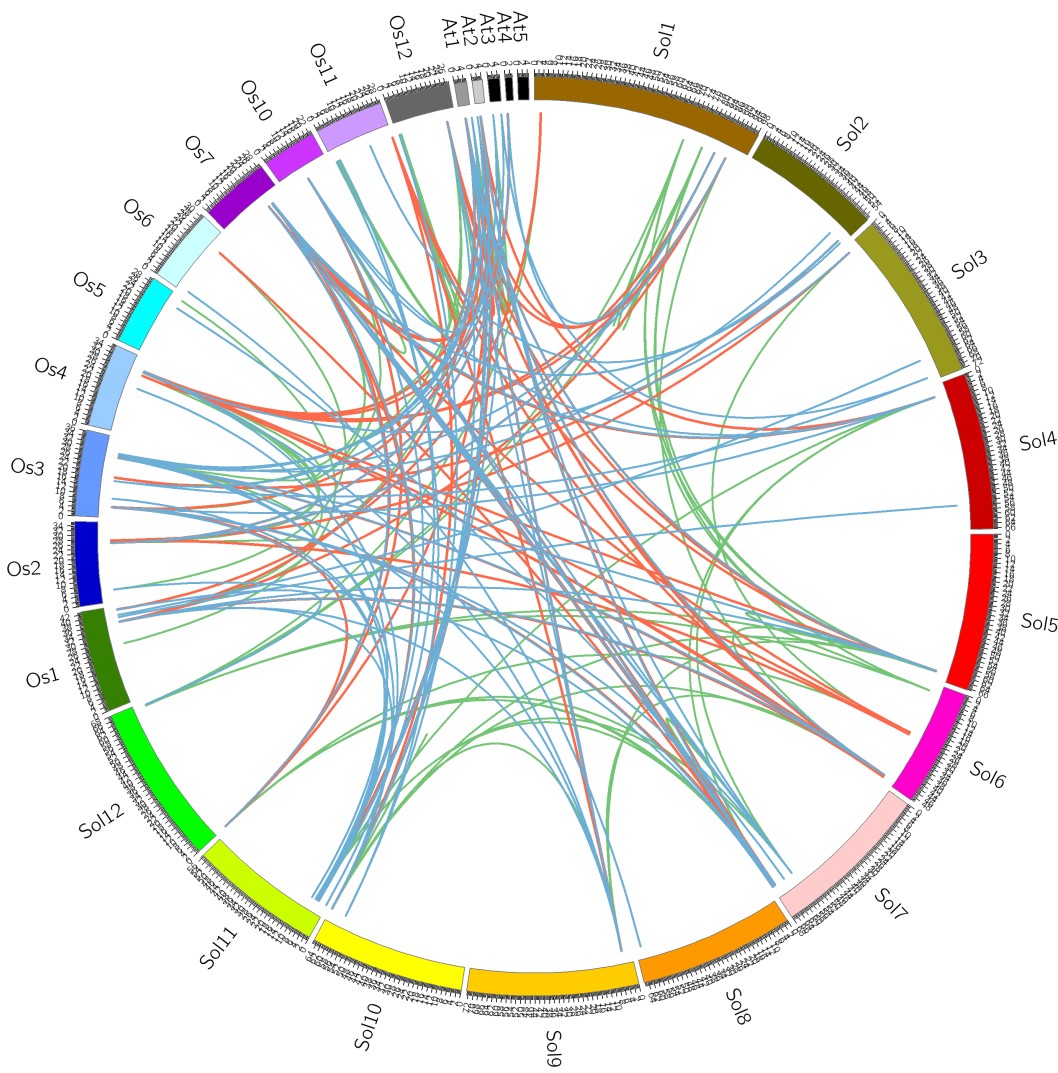

**Figure 3** **Chromosomal localization of GRAS homologous genes in tomato.** Orange, blue and green lines indicate co-orthologous, orthologous and paralogous respectively. The words beginning with A, O and S represent the chromosomes of *Arabidopsis*, rice and tomato, respectively.

that might be due to the absence of temporal expression (*Song et al., 2014*). The other tomato GRAS genes were obtained in at least one tissue. As a whole, we found that the same group of GRAS genes shared a similar expression pattern; for instance, nearly all the PAT1 subfamily members showed a higher expression level than that of other groups, and the Gv6 subfamily showed a low expression level in all tissues. The GRAS genes on some branches exhibited a tissue-specific expression pattern. For example, the *Solyc11g005610.1.1*, *Solyc03g123400.1.1*, and *Solyc09g018460.1.1* genes belonged to the HAM subfamily that showed a low expression in those tissues except the hypocotyls, thereby suggesting that the three genes contributed considerably to the development of the hypocotyl. The *Solyc04g014830.1.1* gene only expresses in the anthesis stage, and thus, would be closely related to the flower opening. *Solyc11g011260.1.1*,

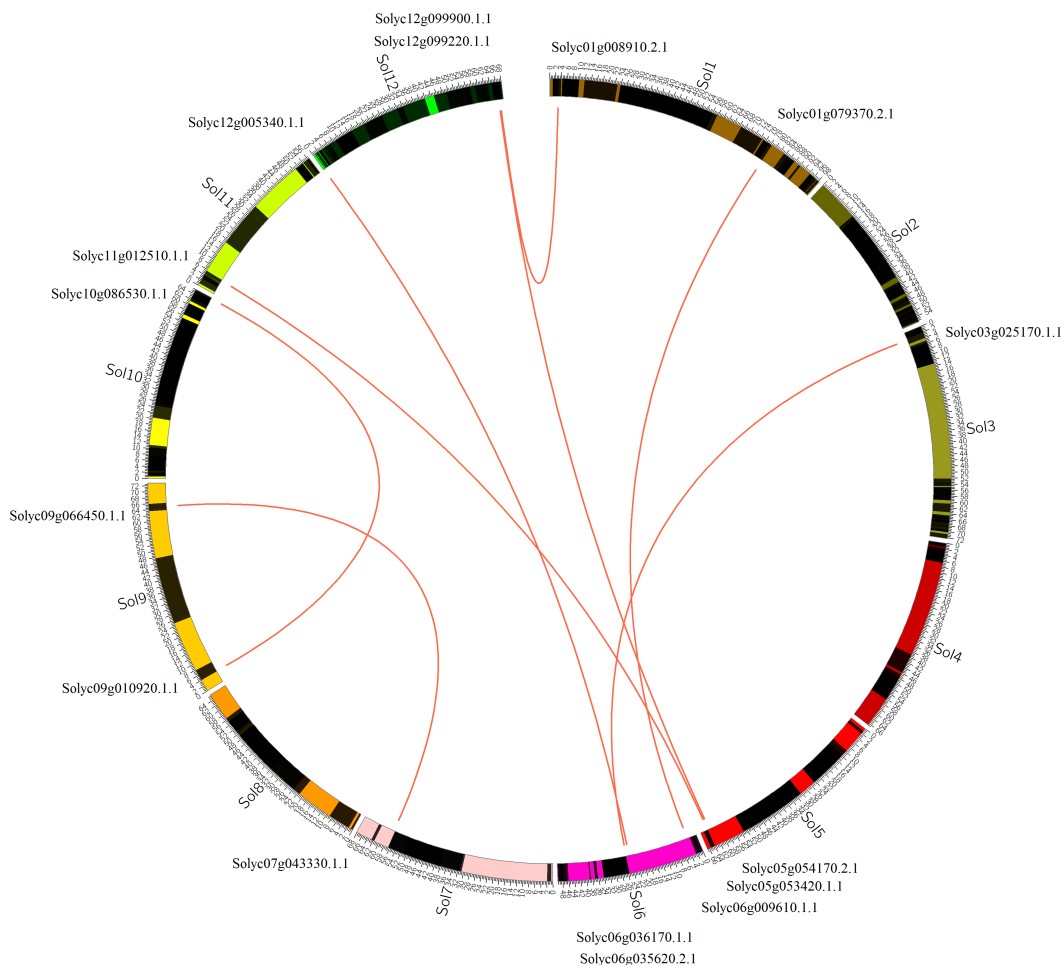

**Figure 4** **Chromosomal localization of GRAS duplicated genes in tomato.** The red lines represent the segmentally duplicated genes, and the black bands represent the collinear block.

*Solyc12g005340.1.1*, and *Solyc11g012510.1.1* genes are always expressed at a high level in all the tissues, indicating their crucial role in the whole growth process of tomato. We also determined that some genes present a time-specific expression, such as *Solyc01g100200.2.1*, *Solyc01g008910.2.1*, *and Solyc07g052960.1.1* which show high expression level in the stage of fruit ripening; the *Solyc10g074680.1.1* gene was expressed in the later stage. During the mature period of leaf and flower, the expression of the GRAS genes remains stable, and only two or three genes show a difference; for example, the *Solyc*11g013150.1.1 gene merely did not express in newly developed leaves, the *Solyc01g.*008910.2.1 only expressed in flowers, and *Solyc02g092570.1.1* only expressed in the newly developed leaves.

According to the public database, using the visualized tool HemI to present the relationship between pstDC3000 and GRAS genes from tomato (Fig. 6), we deduced that the number of higher expression genes in susceptible tomato (54%) were more than that in the resistant tomato (41%). The group of PAT1 continued to show the highest expression
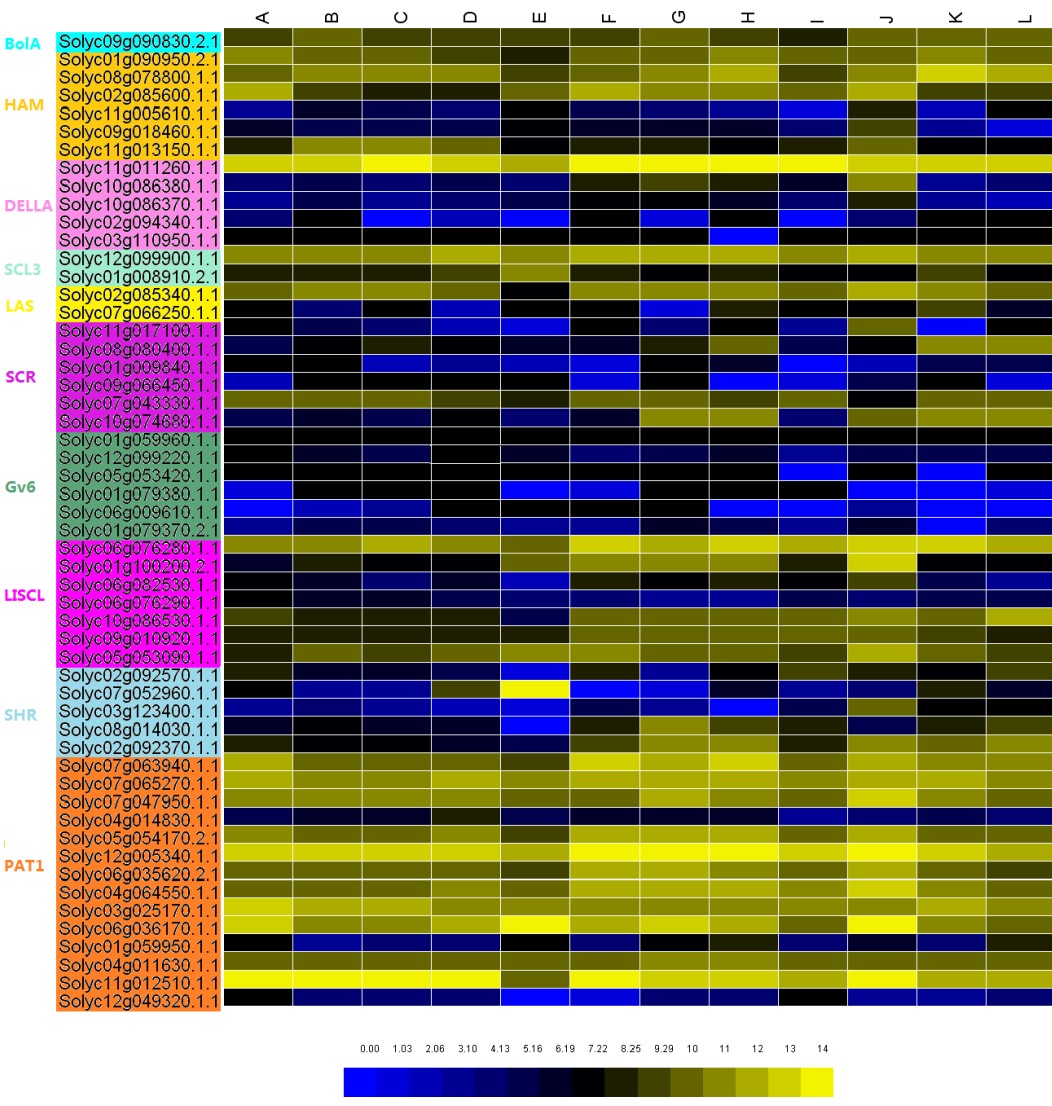

**Figure 5  The expression profile of the tomato GRAS genes in different tissues.** (A) newly developed leaves approximately 5 mm long; (B) mature green leaflets; (C) flower buds 10 days before anthesis or younger; (D) flowers at anthesis; (E) 10 days post anthesis (DPA) fruit; (F) 20 DPA fruit; (G) breaker stage ripening fruit; (H) another set of 10 DPA fruit that was collected in a separate greenhouse for comparison. The following tissues were collected from seeds that were germinated and grown for 7 days in a Petri dish under growing lights: (I) whole root; (J) hypocotyl from below the cotyledons to above the root zone; (K) cotyledons; (L) vegetative meristems and yellow indicating higher expression levels and blue indicating lower expression levels.

level, and the members of the PAT1 subfamily might regulate some critical physiological processes in tomato growth that are related to resistance. During the same infection time, the two varieties of susceptible plant shared the same tendency of the expression of GRAS genes. As the infection time continued, the SISCL group genes' expression was on a downward trend in the resistance tomato, such as *Solyc06g076280.1.1*, *Solyc01g100200.2.2*, and *Solyc06g076290.1.1*.; interestingly, the *Solyc05g053090.1.1* gene was on an increasing

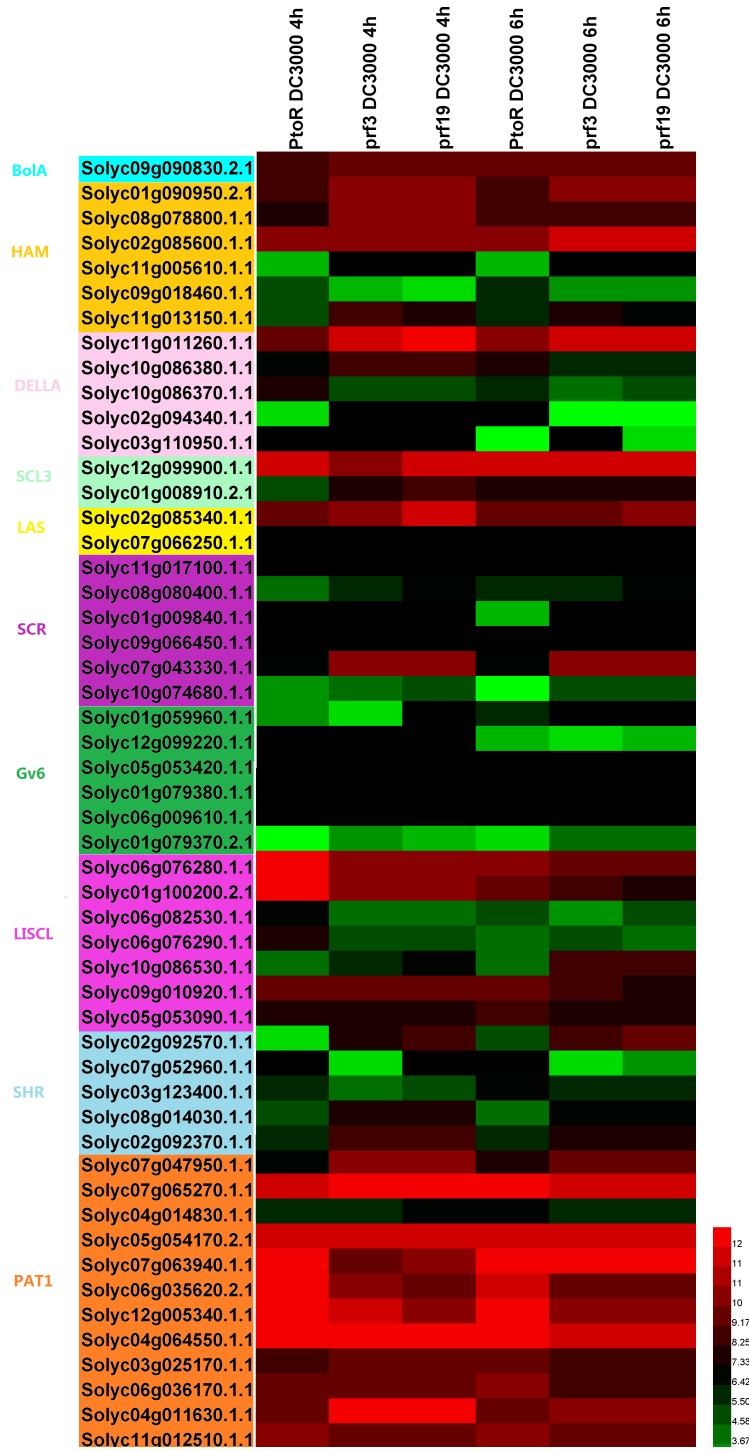

**Figure 6   The expression profile of the tomato GRAS genes in the leaves of resistant (RG-PtoR) and susceptible (RG-prf3 and RG-prf19) tomato plants treated with Pst DC3000.** The red indicates higher expression level and green indicates lower expression levels.

trend. In the susceptible tomato, the *Solyc02g085600.1.1* and *Solyc07g063940.1.1* genes' expressions were increasing, whereas the expression of the other members of the HAM, LISCL, and PAT1 groups were declining. In 4 h, the expression of *Solyc01g100200.2.1*, *Solyc07g063940.1.1*, and *Solyc12g005340.1.1* genes was different between the resistance tomato and the susceptible tomato. In 6 h, the *Solyc12g005340.1.1* gene expression differed largely, and the results suggested that the *Solyc12g005340.1.1* gene regulated the resistance of the plant disease.

## DISCUSSION

Recently, the structural and functional genomics of GRAS transcription factors in higher plant model species have shown that a significant number of members were involved in the plant growth and development (*Heo et al., 2011*), including signal transduction, stress response, meristem formation, cell maintenance, and multiplication. The relevant studies have been elaborately conducted in *Arabidopsis*, which serves as a reference. On the other hand, the GRAS gene characteristics in tomato remain unclear. Thus, we identified 54 GRAS genes from tomato using the bioinformatics methods. Subsequently, the classification and annotation information was obtained, and the full-length sequence of the GRAS proteins from tomato showed remarkable differences. The distribution of GRAS genes in tomato was consistent with that of *Arabidopsis*. The next phylogenetic analysis might provide additional functional constituents among the four species. The previous studies demonstrated that the GRAS transcription factors were involved in plant development and stress response; it is acknowledged that the higher the sequence similarities, the functions were more similar in different species (*Chen et al., 2007*). Additionally, the GRAS proteins have similar functions within the same clade. The structure and domain analysis proved that the topology tree was reliable, and the parallel structural features were clustered to the same subgroup. Every GRAS gene was composed of one or more conserved motifs. A comparative genomics analysis revealed abundant homologous genes in tomato, *Arabidopsis*, and rice, and the segmental duplication commonly promoted the expansion of GRAS proteins (*Cannon et al., 2004*). The expression of GRAS genes in different tissues indicated that the three genes *Solyc11g011260*, *Solyc11g012510*, and *Solyc12g005340* showed a high expression level among those organs, which implicated their vital importance in plant development. Moreover, the members of the same clade shared similar expression profiles (*Wang et al., 2016*). Some of the GRAS genes responded to the biotic stress of pstDC3000; the PAT1 subfamily showed the highest expression level. A similar result was observed in other species for the expression of GRAS genes (*Lu et al., 2015*).

Taken together, the GRAS transcription factor is essential for breeding and cultivation. A total of 54 GRAS members in tomato were identified with respect to gene structures, motifs, and domains; the GRAS proteins showed highly conservative characteristics. A comparative analysis suggested that the functional diversity might be sourced from the large-scale genome duplication. The results of expression of the GRAS genes demonstrated that GRAS transcription factors participate in regulating plant development and responding to biotic/abiotic stress. The present study provided useful information for the functional research in the future.

## Funding

This work was supported by the National Natural Science Foundation of China (Grant No. 31272171), and the Province in Heilongjiang Outstanding Youth Science Fund (No. JC201204). The funders had no role in study design, data collection and analysis, decision to publish, or preparation of the manuscript.

## Grant Disclosures

The following grant information was disclosed by the authors:
National Natural Science Foundation of China: 31272171.
Province in Heilongjiang Outstanding Youth Science Fund: JC201204.

## Competing Interests

The authors declare there are no competing interests.

## Author Contributions

- Yiling Niu conceived and designed the experiments, performed the experiments, analyzed the data, contributed reagents/materials/analysis tools, wrote the paper, prepared figures and/or tables, reviewed drafts of the paper.
- Tingting Zhao, Xiangyang Xu and Jingfu Li contributed reagents/materials/analysis tools, reviewed drafts of the paper.

## Data Availability

The raw data is available as Supplemental Files.

## Supplemental Information

Supplemental information for this article can be found online at http://dx.doi.org/10.7717/peerj.3955#supplemental-information.

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
