# Peer review of "Genome-wide identification and characterization of GRAS transcription factors in tomato (Solanum lycopersicum)"

_PeerJ, doi:10.7717/peerj.3955_

## Round 0.1 · original submission · Major Revisions

In particular please note the comment from reviewer 2 about the Huang et al 2015 paper - please make it clear, both in your manuscript and your response, how this paper differs from that paper and is an advance in the research

Reviewer 1 ·

Basic reporting

'no comment'

Experimental design

'no comment'

Validity of the findings

'no comment'

Additional comments

1. In the part of Materials and Methods, your introduction needs more detail. I suggest that you
illustrate the detailed methods of sampling materials at different stages in order to explain the results at lines 328- 331.
2. In the part of Materials and Methods, please indicate the criteria of excluding acids.
3. The English language should be improved to ensure that your international audience can clearly understand your text. I suggest that you have a native English speaking colleague review your manuscript. Some examples where the language could be improved include lines 291, 296-298, 319,323, 353-358. And some other problems were revised in manuscript.
4. I commend the authors for their extensive data set, compiled analytical works. In addition, the results of manuscript is useful to information for the functional research in the future for tomato. But it is clearly written unambiguous language. If there is a weakness, its English language is not professional in the manuscript (as I have noted above) which should be improved upon before Acceptance.

Annotated reviews are not available for download in order to protect the identity of reviewers who chose to remain anonymous.

Reviewer 2 ·

Basic reporting

The English language should be improved to ensure that your message is transported clearly. I suggest that you have a native English speaking colleague review your manuscript.
Some examples for the beginning of the Ms:
rational root patterning - radial
(42)The transcription factors are essential in functional genomics - reword
(47)The GRAS transcription factor is - you are talking about a gene family
(65) The functional localization of the motif - reword
(68) In addition, the N-terminal region of GRAS proteins in the DELLA subfamily contains
the other two conserved motifs ......., indicating that the N-terminal domains of GRAS proteins are variable - finding motives does not necessarily make them variable
(83) Hitherto, the RNA-seq is developed, following the function of GRAS genes, which are
deeply excavated - can RNA-seq lead to function?
(90) the inhibition of DELLA proteins was released immediately by the 26S proteasome - the degradation by the DELLA protein through the proteasome released the inhibition

Make sure all spaces are at the correct position and check for italic on the plant names.
(49) Arabidopsis(Arabidopsisthaliana) first Arabidopsis not italic.
I think Brassicaceae is more commonly used in contrast to Cruciferae.
Recheck the gene names: e.g. 265 Solyc090830.2.2
Some newer literature references on GRAS proteins seem to be missing. Especially the discussion on the evolvement from methyltransferases could be important for the analysis of conserved motives.
Fig. 1 (and others) - please explain colour coding and add some more information to the legend.
Fig. 2 - might be too small writing for printing

As this is an interesting field and genome wide analysis could tell us much about understanding evolution and the role of these factors in different developmental stages this could be more clarified in the discussion.

Experimental design

GRAS sequences were extracted from Solanum lycopersicum but the expression data were from S. pimpinellifolium? Why were those uesed and is there a difference between the GRAS genes in both species?
(205) in some GRAS proteins were less than that in the typical GRAS
domain, such as Solyc04g011630.1.1 or Solyc06g076290.1.1genes....
Did the authors check if this is just an annotation problem or if these genes might be pseudo genes?

Validity of the findings

(220) Strikingly, some clades do not contain the members of GRAS proteins from Arabidopsis or rice; for instance, Gv6 and BolA groups. This might be attributed to the loss of the corresponding member following the separation of the last common ancestor. - is there any prove that it is a loss of this branch not a gain?

(320) We also determined that some genes present a time-specific expression, such as, in the stage of fruit ripening; several genes are expressed in the later stage. - This is an important finding and should be elaborated upon. Fig. 5, where this data is derived from, is not very easy to follow - perhaps some symbols of grouping to different stages whould make this easier to read and interprete. Are the genes listed according to their branch (that could be added for clarification) or was there any clustering performed?
As you state "As a whole, we found that the same group of GRAS genes shared a similar expression pattern..." this is not compehensible from the figure at first glance.
Confimation of these data via qPCR might be recommendable.

In 2015 Huang et al. have put together a genome-wide analysis of GRAS proteins in tomato and added also expression data. The authors should make it more obvious in how this study differs from the previous study.

---

## Round 0.2 · Minor Revisions

Whilst you have addressed most of the reviewers' comments the English still requires substantial revision. It needs to be edited by a native English speaker. There are also many typographical errors which need to be thoroughly checked, for example line 45 rice has the incorrect name (it should be Oryza sativa).

---

## Round 0.3 · Minor Revisions

Whilst you have improved the manuscript, it still requires English language editing. I suggest you get a native English speaker to go through it and edit it thoroughly.

---

## Round 0.4 · accepted · Accept

Thank you for making the necessary corrections, the paper is now acceptable for publication